# Effective Equilibrium in Out-of-Equilibrium Interacting Coupled Nanoconductors

**DOI:** 10.3390/e22010008

**Published:** 2019-12-19

**Authors:** Lucas Maisel, Rosa López

**Affiliations:** Institut de Física Interdisciplinària i de Sistemes Complexos IFISC (CSIC-UIB), E-07122 Palma de Mallorca, Spain; lucas.maisel@gmail.com

**Keywords:** quantum transport, quantum dots, fluctuation–dissipation theorem, Onsager relations

## Abstract

In the present work, we study a mesoscopic system consisting of a double quantum dot in which both quantum dots or artificial atoms are electrostatically coupled. Each dot is additionally tunnel coupled to two electronic reservoirs and driven far from equilibrium by external voltage differences. Our objective is to find configurations of these biases such that the current through one of the dots vanishes. In this situation, the validity of the fluctuation–dissipation theorem and Onsager’s reciprocity relations has been established. In our analysis, we employ a master equation formalism for a minimum model of four charge states, and limit ourselves to the sequential tunneling regime. We numerically study those configurations far from equilibrium for which we obtain a stalling current. In this scenario, we explicitly verify the fluctuation–dissipation theorem, as well as Onsager’s reciprocity relations, which are originally formulated for systems in which quantum transport takes place in the linear regime.

## 1. Introduction

The two paradigms of statistical mechanics for systems that are close to equilibrium are: (i) the Onsager–Casimir reciprocity relations [1]; and (ii) the fluctuation–dissipation theorem (FDT) [2,3,4]. Both relations are not only inherent to classical systems but are also applicable to the quantum regime. The Onsager–Casimir reciprocity relations state that the Onsager matrix that relates physical fluxes and their conjugate forces is symmetric. For example, considering as forces the electrostatic and thermal gradients, and their associated currents being the electrical and heat fluxes, these relations set an identity between the thermoelectrical conductance (electrical response to a thermal gradient) and the electrothermal conductance (response of the heat current to an electrical bias). On the other hand, the FDT establishes that statistical fluctuations occurring in a system at equilibrium behave similarly to the dissipation that takes place under the action of an external perturbation. Major examples of manifestations of the FDT are found in Einstein’s treatment of Brownian motion where the diffusion constant is found to be proportional to the mobility [5] or the Johnson–Nyquist formula for electronic white noise [6]. In the context of quantum transport through electronic nanodevices, the FDT allows us to relate the dissipative response of one current with respect to a variation of its affinity or conjugate force with its spontaneous fluctuations. This property of equilibrium systems is a very important topic when we are interested in controlling dissipation due to currents induced through quantum conductors by external forces.

As mentioned above, the range of validity of the FDT is limited to the linear response regime, i.e., for sufficiently small perturbations. Going beyond this regime requires generalizing this formulation to non-equilibrium conditions. This has been done by introducing additional correlations involving the activity, a magnitude related to the transition rates and the excess of entropy production that is modified antisymmetrically by the external potential that drives the system out of equilibrium [7,8,9,10]. In this view, these extensions to the FDT are indeed fluctuation–dissipation relations (FDR) that establish the frequency at which a system produces entropy to the environment between forward and backward processes. The interest of the FDR has been highlighted in the field of quantum transport [1,11,12,13].

However, here we adopt a different perspective, reported in the work of B. Altaner, M. Polettini, and M. Esposito [14], in which the concept of *stalling currents* is introduced in the context of stochastic thermodynamics. A current that traverses a system can be nullified because of the cancellation of a set of distinct internal processes, and is then called a *stalling current*. Under these conditions, if the perturbative force solely affects the microscopic transitions that contribute to this current, the FDT is restored [14,15]. In addition, we test numerically that Onsager reciprocity relations are additionally satisfied at stalling conditions. We speculate that this property is attained due to the lack of entropy production at stalling conditions forced by the tight coupling between the charge and heat currents (see below). The conclusion is that all contributing elemental transitions being internally equilibrated is equivalent to them being microscopically reversible. One interesting application to the stalling configuration is that, even though correlations are usually difficult to access experimentally, the fact that the FDT is applicable makes it rather easy to obtain such correlations by means of a response function instead.

Our purpose in this work is to implement these conclusions in a nanodevice consisting of two interacting conductors. Such setup was previously investigated by R. Sánchez et al. [16] to analyze the drag effect. The device consists of a parallel double quantum dot system in which the quantum dots interact electrostatically via a mutual capacitance. Besides, each quantum dot is tunnel-connected to two electronic reservoirs. A drag current is encountered in one of the dots, which is unbiased, due to the charge fluctuations provoked by the electrical current driven through the other dot. The detection of this drag current has been demonstrated experimentally [17] showing that high-order tunneling events such as cotunneling have a significant contribution. Besides, a drag current control has been proposed by attaching to the dots different materials with nontrivial energy-dispersion relations [18]. This system has additionally been proposed for the implementation of a Maxwell demon [19], in which one of the dots (the demon) acquires information from the other one, allowing a current to flow opposite to the applied bias voltage in the other dot.

Our goal in this article is to explore the transport properties in an out-of-equilibrium configuration that drives the system into an effective equilibrium in which both the Onsager relations and the FDT are recovered. For this purpose, we compute the electrical and heat currents through each quantum dot. By a numerical search of stalling currents in one of the dots, we check whether or not Onsager relations and the FDT are satisfied. We consider different situations. Firstly, we consider the case where both the electrical and heat flows are cancelled simultaneously under non-equilibrium configurations. This can be achieved only in the so-called strong coupling regime. For this case, we demonstrate that the system indeed behaves as at equilibrium. We also analyze the scenario where only one of the currents vanishes (either the charge or the heat flow), while the other one is kept finite. Finally, we show that the absence of stalling currents prevents the fulfillment of the Onsager relations and the FDT as expected. To conclude, we go beyond the FDT and additionally check the FDRs for the third cumulant in the presence of stalling currents.

## 2. Theoretical Model

### 2.1. Description of the System and Underlying Framework

We consider the case of two conductors that are mutually connected via the Coulomb interaction. Each conductor consists of a quantum dot with a single level active for transport. We omit spin indices due to spin degeneracy. Besides, we consider a large on-site Coulomb interaction that prevents the double occupancy in each dot. Each quantum dot is tunnel-coupled to two electronic reservoirs that can be biased with electrostatic and thermal gradients. Each tunneling barrier is modeled by capacitors denoted by $C_{i}$ with i=1,2,3,4. As mentioned above, the two quantum dots interact electrostatically through a capacitor *C*. A sketch for this system is depicted in Figure 1b. Under these circumstances, we describe the system using four possible charge states $\left|0\rangle =\right|0_{u}0_{d}\rangle$, $\left|u\rangle =\right|1_{u}0_{d}\rangle$, $\left|d\rangle =\right|0_{u}1_{d}\rangle$, and $\left|2\rangle =\right|1_{u}1_{d}\rangle$, where $n_{u}n_{d}$ denotes the charge state with $n_{u}$ electrons in the upper dot and $n_{d}$ electrons in the lower dot. For simplicity, we consider an isothermal configuration in which all reservoirs are held at a common temperature *T*. We also keep different bias voltages $V_{i}$ applied to the four terminals.

We are interested in the charge and heat transport in the sequential tunneling regime, in which the tunneling rate (denoted by Γ) satisfies $\hbar \Gamma \ll k_{B}T$. In this regime, transport of electrons along each quantum dot occurs in a sequence of one electron transfer event at each time. Electrons can hop into a quantum dot, and then relax before they jump again. This restriction eliminates the transitions |0(2)〉→|2(0)〉 and |u(d)〉→|d(u)〉. Additionally, we consider that there is no particle transfer from one dot to another by tunneling. The only interaction between the dots is then due to their mutual influence caused by the electrostatic interactions.

The theoretical framework employed to describe the quantum transport in our system is called *stochastic thermodynamics* [14,20,21,22]. Quite generally, we can consider a setup with an arbitrary number of states n∈{1,2,…,N} and picture each state as a node in a connected network. We draw edges **e** connecting states between which a transition may occur, and require these to be possible in both directions. However, transitions along ±**e** are not required to happen at the same rate or with the same probability. Note that two nodes may be connected with several edges if there are various physical mechanisms through which the system can transition between the associated states. The evolution of the system is modeled as a Markov jump process, i.e., the probability that the system jumps from one state to another is independent of its previous history. This evolution can also be visualized as a random walk on the network. A physical model is defined by prescribing the forward and backward transition rates $w_{\pm e}$, which evidently may be functions of the physical parameters involved. The fluctuating current along an edge **e**, $j_{e}\left(t\right)=\sum_{k}\delta \left(t-t_{k}\right)\left(\delta_{+e,e_{k}}-\delta_{-e,e_{k}}\right)$, is a stochastic variable that peaks if the system transitions along the directed edge $e_{k}$ at time $t_{k}$. Physical currents, i.e., currents associated to the transport of physical quantities such as charge or heat, are weighted currents $J_{\alpha }=\sum_{e}d_{e}^{\alpha }j_{e}$, where $d_{+e}^{\alpha }=-d_{-e}^{\alpha }$ specifies the amount of a physical variable α exchanged with an external reservoir along a transition edge e.

When applying the previous theoretical treatment to our particular system, we consider that the tunneling rates depend on the energy of the system. Specifically, we consider the value of $\Gamma_{i}$ for the tunneling of electrons between a reservoir *i* and a quantum dot whenever the other dot is empty, and $\gamma_{i}$ when the other dot is occupied. Then, the transition rates (previously called $w_{\pm e}$) are thus dependent on the dot charge states. The transition rates are defined according to Fermi’s golden rule as
(1)$$\begin{array}{cc} \Gamma_{i}^{-} & =\Gamma_{i}f\left(\mu_{\ell ,0}-qV_{i}\right) \end{array}$$
(2)$$\begin{array}{cc} \Gamma_{i}^{+} & =\Gamma_{i}\left(1-f(\mu_{\ell ,0}-qV_{i})\right) \end{array}$$
(3)$$\begin{array}{cc} \gamma_{i}^{-} & =\gamma_{i}f\left(\mu_{\ell ,1}-qV_{i}\right) \end{array}$$
(4)$$\begin{array}{cc} \gamma_{i}^{+} & =\gamma_{i}\left(1-f(\mu_{\ell ,1}-qV_{i})\right) \end{array}$$
where $f\left(x\right)={\left(1+e^{x/k_{B}T}\right)}^{-1}$ is the Fermi–Dirac distribution function, the − superscript stands for the tunneling from the lead to the dot, and + for the reverse process. The transition rates are schematically represented in a network diagram in Figure 2. In our arrangement, we take the *up* dot (ℓ=u) connected to left and right reservoirs with i={1,2}, and the *down* dot (ℓ=d) to reservoirs with i={3,4}. Note that the numerical subindex in the previous transition rates thus indicates the reservoir involved in the transition, as shown in Figure 1a. The chemical potential for the dot *ℓ*, i.e., $\mu_{\ell ,0}$ ($\mu_{\ell ,1}$), corresponds to the situation in which the *other* dot is empty (occupied).

To determine the effective chemical potentials of the dots, we must develop a model that takes into account how their energy levels are influenced by electrostatic interactions. When interactions are properly included as in our description, all currents are gauge invariant, as they depend only on voltage differences. Hereafter, we shorten the notation and define $V_{ij}\equiv V_{i}-V_{j}$. Under these considerations, the dot levels become
(5)$$\begin{array}{c} \varepsilon_{u,n}\to \mu_{u,n}=\varepsilon_{u}+U\left(1,0\right)-U\left(0,0\right)+E_{C}\delta_{1n} \end{array}$$
(6)$$\begin{array}{c} \varepsilon_{d,n}\to \mu_{d,n}=\varepsilon_{d}+U\left(0,1\right)-U\left(0,0\right)+E_{C}\delta_{1n} \end{array}$$
where $\varepsilon_{u}$ and $\varepsilon_{d}$ are the bare energy levels, and n=0(1) corresponds to the case where *other* dot is empty (occupied). Here, $E_{C}=2q^{2}C/\left(C_{\Sigma u}C_{\Sigma d}-C^{2}\right)$ is the charging energy with $C_{\Sigma d(u)}=C_{1(3)}+C_{2(4)}+C$. The chemical potential $\mu_{u(d),n}$ is defined as the change in the electrostatic energy when the charge number $N_{u(d)}$ changes by one when the dot d(u) is either empty (n=0) or occupied by one electron (n=1). The electrostatic energy is computed from $U\left(N_{u},N_{d}\right)=\sum_{i}\int_{0}^{qN_{i}}dQ_{i}^{\prime }\;\phi_{i}\left(Q_{i}^{\prime }\right)$ where $\phi_{i}$ is the internal potential in each quantum dot obtained by means of elementary electrostatic relations. Then, the arguments of the Fermi functions appearing in the tunneling rates read [16]:(7)$$\begin{array}{cc} \mu_{u,n}-qV_{1} & =\varepsilon_{u}+\frac{1}{C_{\Sigma u}C_{\Sigma d}-C^{2}}\left[\frac{q^{2}}{2}C_{\Sigma d}+q\left(C_{\Sigma d}C_{2}V_{21}+CC_{3}V_{31}+CC_{4}V_{41}\right)\right]+E_{C}\delta_{1n} \end{array}$$(8)$$\begin{array}{cc} \mu_{u,n}-qV_{2} & =\varepsilon_{u}+\frac{1}{C_{\Sigma u}C_{\Sigma d}-C^{2}}\left[\frac{q^{2}}{2}C_{\Sigma d}+q\left(C_{\Sigma d}C_{1}V_{12}+CC_{3}V_{32}+CC_{4}V_{42}\right)\right]+E_{C}\delta_{1n} \end{array}$$(9)$$\begin{array}{cc} \mu_{d,n}-qV_{3} & =\varepsilon_{d}+\frac{1}{C_{\Sigma u}C_{\Sigma d}-C^{2}}\left[\frac{q^{2}}{2}C_{\Sigma u}+q\left(C_{\Sigma u}C_{4}V_{43}+CC_{1}V_{13}+CC_{2}V_{23}\right)\right]+E_{C}\delta_{1n} \end{array}$$(10)$$\begin{array}{cc} \mu_{d,n}-qV_{4} & =\varepsilon_{d}+\frac{1}{C_{\Sigma u}C_{\Sigma d}-C^{2}}\left[\frac{q^{2}}{2}C_{\Sigma u}+q\left(C_{\Sigma u}C_{3}V_{34}+CC_{1}V_{14}+CC_{2}V_{24}\right)\right]+E_{C}\delta_{1n} \end{array}$$
that now depend only on voltage differences. The four $\mu_{u,n}-V_{1(2)}$, and $\mu_{d,n}-V_{3(4)}$ are the electrochemical potentials. We take $V_{12}$, $V_{13}$ and $V_{34}$ as the only independent biases, since the rest of voltage differences can be expressed as linear combinations of these values.

As discussed above, we apply the Markov approximation in order to determine the dynamics of the probabilities of finding the system in one of the four states. Specifically, we employ the master equation formalism, where the time evolution of the system is governed by a master equation that gives the probability distribution of the considered stochastic variables in terms of the transition rates between the different states. Defining $\Gamma_{u(d)}^{\pm }=\Gamma_{1(3)}^{\pm }+\Gamma_{2(4)}^{\pm }$, the following relations are found:(11)$$\left(\begin{array}{c} \dot{p_{0}}\\ \dot{p_{u}}\\ \dot{p_{d}}\\ \dot{p_{2}} \end{array}\right)=\left(\begin{array}{cccc} -\Gamma_{u}^{-}-\Gamma_{d}^{-} & \Gamma_{u}^{+} & \Gamma_{d}^{+} & 0\\ \Gamma_{u}^{-} & -\Gamma_{u}^{+}-\gamma_{d}^{-} & 0 & \gamma_{d}^{+}\\ \Gamma_{d}^{-} & 0 & -\gamma_{u}^{-}-\Gamma_{d}^{+} & \gamma_{u}^{+}\\ 0 & \gamma_{d}^{-} & \gamma_{u}^{-} & -\gamma_{u}^{+}-\gamma_{d}^{+} \end{array}\right)\left(\begin{array}{c} p_{0}\\ p_{u}\\ p_{d}\\ p_{2} \end{array}\right)$$As we are interested in the steady state, we set all $\dot{p_{i}}=0$. Considering the normalization condition $\sum_{i}p_{i}=1$, we obtain:(12)$$\begin{array}{cc} p_{0} & =\frac{1}{\alpha }\left[\Gamma_{d}^{+}\gamma_{u}^{+}\left(\Gamma_{u}^{+}+\gamma_{d}^{-}\right)+\Gamma_{u}^{+}\gamma_{d}^{+}\left(\Gamma_{d}^{+}+\gamma_{u}^{-}\right)\right] \end{array}$$(13)$$\begin{array}{cc} p_{u} & =\frac{1}{\alpha }\left[\Gamma_{u}^{-}\Gamma_{d}^{+}\left(\gamma_{u}^{+}+\gamma_{d}^{+}\right)+\gamma_{u}^{-}\gamma_{d}^{+}\left(\Gamma_{u}^{-}+\Gamma_{d}^{-}\right)\right] \end{array}$$(14)$$\begin{array}{cc} p_{d} & =\frac{1}{\alpha }\left[\Gamma_{u}^{+}\Gamma_{d}^{-}\left(\gamma_{u}^{+}+\gamma_{d}^{+}\right)+\gamma_{u}^{+}\gamma_{d}^{-}\left(\Gamma_{u}^{-}+\Gamma_{d}^{-}\right)\right] \end{array}$$(15)$$\begin{array}{cc} p_{2} & =\frac{1}{\alpha }\left[\gamma_{u}^{-}\gamma_{d}^{-}\left(\Gamma_{u}^{-}+\Gamma_{d}^{-}\right)+\Gamma_{u}^{-}\Gamma_{d}^{+}\gamma_{d}^{-}+\Gamma_{u}^{+}\Gamma_{d}^{-}\gamma_{u}^{-}\right] \end{array}$$
with
(16)α=Γu−Γd+γu++γd++γd−γu+Γd++Γu+Γd+γu++γd++Γdγd−γu++Γuγd+γu−+=+γd−Γu+γu+γd++γu−γd
and $\Gamma_{u(d)}=\Gamma_{u(d)}^{+}+\Gamma_{u(d)}^{-}$ (similar for $\gamma_{u(d)}$).

We now compute the electrical current $I_{1}$ that flows between the first lead and the upper dot, which, we from now on call *drag current* for historical reasons (note that since generally $V_{12}\neq 0$ it is not a current arising solely from the drag effect). This current is obtained by weighting the transition probabilities with the electron charge *q*. The result is
(17)$$I_{1}=q\left(\Gamma_{1}^{-}p_{0}-\Gamma_{1}^{+}p_{u}+\gamma_{1}^{-}p_{d}-\gamma_{1}^{+}p_{2}\right)$$Because of electric charge conservation, we immediately know $I_{2}=-I_{1}=I$ for the current between the second terminal and the *up* dot (we assign a + sign whenever the current flows from a lead into a dot, and a − sign otherwise). We can also compute the heat current by weighting the transitions with the amount of transferred effective energy (the electrochemical potential),
(18)$$J_{1}={\tilde{\mu }}_{u,0}\left(\Gamma_{1}^{-}p_{0}-\Gamma_{1}^{+}p_{u}\right)+{\tilde{\mu }}_{u,1}\left(\gamma_{1}^{-}p_{d}-\gamma_{1}^{+}p_{2}\right)$$
where ${\tilde{\mu }}_{u,n}=\mu_{u,n}-qV_{1}$. Similar expressions are obtained for the rest of the $J_{i}$. Energy conservation leads to $J_{1}+J_{2}+J_{3}+J_{4}=I_{1}V_{21}+I_{3}V_{43}$. These currents were investigated by Sánchez et al. [16] when the up dot is at equilibrium with $V_{1}=V_{2}$; a nonzero drag current $I_{1}$ then appears when $\Gamma_{1}\gamma_{2}$≠$\gamma_{1}\Gamma_{2}$. This means that the current in the lower terminals (drive system) drives the upper dot towards a non-equilibrium situation by the appearance of a drag current. The drag phenomenon can be clearly understood from the following Joule relation found for this setup
(19)$$\begin{array}{ccc} J_{1}+J_{2}+J_{c} & = & -I_{1}V_{12}\\ J_{3}+J_{4}-J_{c} & = & -I_{3}V_{34} \end{array}$$
where
(20)$$J_{c}=\frac{E_{C}}{\alpha }\left(\gamma_{u}^{+}\gamma_{d}^{-}\Gamma_{u}^{-}\Gamma_{d}^{+}-\gamma_{u}^{-}\gamma_{d}^{+}\Gamma_{u}^{+}\Gamma_{d}^{-}\right)$$
is the heat flow between the drag and the drive system. This expression generalizes the relation found in Reference [23] for a three-terminal double quantum dot. In such a system, the drag conductor is connected to two reservoirs, whereas the drive dot is coupled to a single contact. Therefore, the drive subsystem does not support any charge current. Under these considerations, the drive dot carries a heat flow *J* (with a similar form to Equation (Equation 20), which is proportional to the drag charge current when $\Gamma_{2}=\gamma_{1}=0$, i.e., in the so-called strong coupling regime. Here, Equation (Equation 19) demonstrates the existence of a heat flow $J_{c}$ between the drag and drive subsystems. This energy flow appears in addition to the heat flows $J_{1}$, $J_{2}$ through the drag conductor and the heat currents $J_{3}$, $J_{4}$ in the drive subsystem, even when they are held at common temperature and no particle transfer exists between them.

Returning to our purpose, which is to find a route to an effective equilibrium state, we address the issue of whether the opposite phenomenon to the drag is possible, i.e., if we can achieve a non-equilibrium configuration with $V_{1}\neq V_{2}$ for which the drag effect causes the stalling of the upper currents. Under this novel situation, we check whether our system reaches an “effective linear response regime” by testing the microreversibility property through the Onsager relations and the fulfilment of the FDT. To this end, we focus on the *up* dot and consider three stalling configurations: (i) when both the electrical and the heat flow vanish, i.e., $I_{1}=0$ and $J_{1}=0$, which we call the globally stalled scenario; (ii) when the charge current is nullified, $I_{1}=0$, but there is a finite heat flow $J_{1}\neq 0$; and (iii) when there is a finite electrical current $I_{1}\neq 0$ but no heat flow, $J_{1}=0$. These situations correspond to the locally charge-stalled and heat-stalled cases, respectively. The simplest manner to achieve the globally stalled case is tuning the system to the strong coupling configuration by setting $\gamma_{1}=\Gamma_{2}=0$. Under this situation, electrons can only tunnel in and out of the top-left reservoir if the lower dot is empty, and of the top-right reservoir if the lower dot is occupied.

### 2.2. Detailed Balance and Behavior at Equilibrium

Before presenting our results, we carefully revise the behavior of systems near thermodynamic equilibrium. In this situation, all existing currents in a system tend to zero on average. This behavior is called *global detailed balance*. According to statistical mechanics, systems subject to these conditions exhibit the property that the correlations of the spontaneous fluctuations and the dissipative response to an external perturbation obey the same rules, which is primarily known as Onsager’s regression hypothesis [14]. This important statement is the heart of the fluctuation–dissipation theorem (FDT). If we consider an arbitrary physical current $J_{\alpha }$ (such as a heat or charge current) and its affinity or conjugate force $h_{\alpha }$ (which in these cases would correspond to gradients in temperature or electrical potential, respectively), the theorem can be expressed as
(21)$$\partial_{h_{\alpha }}J_{\alpha }\left(x^{eq}\right)=D_{\alpha ,\alpha }\left(x^{eq}\right)$$
where $D_{\alpha ,\alpha }$ is a generalized diffusion constant proportional to $\langle J_{\alpha }J_{\alpha }\rangle$. The vector x contains all the parameters the current may depend on, and satisfies $J_{\alpha }\left(x^{eq}\right)=0$ for all currents in the system; their conjugate forces are evidently also required to vanish. The previous equation can be generalized in such a way that it expresses the FDT for the combination of two currents and their conjugate forces by changing one index α for a different one and symmetrizing both sides of the expression (see complementary material of Reference [14]).

Another major result in thermodynamics close to equilibrium is found in Onsager’s reciprocal relations (RRs), which actually follow from the FDT if the system enjoys the property of being time-reversible [15]. In the following, we restrict ourselves to relations between heat and charge currents, following Onsager’s original article [1]. For a system where transport of these quantities exists, the mechanisms are usually not independent, but interfere with each other leading to the well known thermoelectric effects. If we consider a system at equilibrium, small fluctuations or external perturbations may allow for the transport of small quantities of charge and heat while the system is returning to its original state. Onsager established that, in these situations, the responses of a current due to a variation of the other current’s conjugate force are equal, i.e., the heat current responds in the same way to a variation of the electrical potential as the charge current to a temperature fluctuation. This result is best visualized by writing the currents in matrix form. For a simple system with a single heat and charge current, we have:(22)$$\left(\begin{array}{c} J_{charge}\\ J_{heat} \end{array}\right)=\left(\begin{array}{cc} L_{11} & L_{12}\\ L_{21} & L_{22} \end{array}\right)\left(\begin{array}{c} \delta \left(\Delta V\right)\\ \delta \left(\Delta T/T\right) \end{array}\right)$$
where $L_{11}$ and $L_{22}$ are the electrical and thermal conductances, and $L_{12}=\partial_{\Delta T/T}J_{charge}$ and $L_{21}=\partial_{\Delta V}J_{heat}$ represent the electrothermal and thermoelectrical coefficients that arise from the interference of the two transport mechanisms. Onsager’s statement is then equivalent to the requirement that the conductance matrix be symmetric, $L_{12}=L_{12}$. In addition to these relations, the scattering theory formalism ensures that both the thermal and the electrical conductances are semipositive.

Despite these theorems being major cornerstones in our understanding of the behavior of systems obeying global detailed balance, most complex systems live out of equilibrium. Accordingly, similar relations have been sought for systems where detailed balance is explicitly broken, since their finding would allow us to characterize and study out-of-equilibrium systems in a similar manner as when detailed balance is satisfied.

### 2.3. Local Detailed Balance and Equilibrium-Like Relations

A central assumption in stochastic thermodynamics far from equilibrium, when global detailed balance is not satisfied, is local detailed balance (LDB). It relates the forward and backward transition rates *w* into and out of a state *A* by means of a mechanism ν and reads [24]
(23)$$\frac{w_{A\to B}^{\nu }}{w_{B\to A}^{\nu }}=e^{-\beta_{\nu }\Delta \varepsilon }$$
where $\beta_{\nu }$ is the inverse temperature of the reservoir involved in the transition and Δε is the difference between the energies of states *A* and *B*. It can be easily checked that the rates in Equations (Equation 1)–(4) indeed satisfy the LDB condition.

In Reference [14], it was reported that, if LDB is satisfied in a system driven arbitrarily far from equilibrium, its response to a perturbation or a spontaneous fluctuation may obey a relation similar to the equilibrium FDT if certain additional conditions are fulfilled. More precisely, it has been established that a current $J_{\alpha }$ in such a system obeys Equation (Equation 21) with $x^{eq}$ replaced by $x^{st}$, where $x^{st}$ corresponds to a configuration of the parameters of the current such that $J_{\alpha }\left(x^{st}\right)=0$, i.e., the considered current stalls. This is valid if the force $h_{\alpha }$ couples exclusively to those transitions that contribute to the conjugate current $J_{\alpha }$. It is important to notice the difference between this statement and the first FDT valid only near equilibrium, since we now only require a given current to stall internally. This may be a consequence of the appropriate tuning of the rest of the currents in the system, which are no longer required to vanish, and can in fact assume arbitrary magnitudes.

Similarly, Onsager’s reciprocal relations have also been extended to non-equilibrium situations, under the condition of a marginal time-reversibility [25]. Again, it is required that the currents stall in order for the RRs to hold far from equilibrium.

## 3. Results and Discussion

In this section, we present the main results of our work. We verify the RRs and the FDT for a complete understanding of the impact of stalling currents in coupled conductors.

### Roots of the Drag Current and Equilibrium-Like Behavior

The aim of our study is to verify the generalized non-equilibrium reciprocity relations and the fluctuation–dissipation relations. As discussed above, they require that the involved currents be at stall in order to hold arbitrarily far from equilibrium. We exclusively focus on situations where the stalling currents are those between the upper dot and the first lead, i.e., the ones in the drag system. Since $I_{1}=-I_{2}$, it is enough for our purposes to seek for roots of $I_{1}$. We also only look for roots of $J_{1}$, even though $J_{1}\neq J_{2}$. For all the out-of-equilibrium calculations, we consider the isothermal case $T=T_{i}$, with i=1,…,4. Since we are only interested in the responses of the currents to small temperature fluctuations in one of the leads (with the rest held constant), we must formally treat the temperatures in each lead as independent of each other for computational means. However, in the end, all derivatives are evaluated at temperature *T*.

The electric current $I_{1}$ [Equation (Equation 17)] is a highly nonlinear function of the biases $V_{12}$, $V_{13}$ and $V_{34}$. Consequently, the solutions to $I_{1}=0$ must be found by means of numerical analysis in order to verify Onsager’s relations and the FDT (further justifications below). To this purpose, we set $\Gamma_{i}=\gamma_{i}=\Gamma$ except for $\gamma_{1}=0.1\Gamma$, $k_{B}T=5\hbar \Gamma$, $q^{2}/C_{i}=20\hbar \Gamma$, $q^{2}/C=50\hbar \Gamma$ and $\varepsilon_{u}=\varepsilon_{d}=0$. Furthermore, we consider natural units where $\hbar =-q=k_{B}=\Gamma =1$. Unless otherwise mentioned, these parameters are used in the rest of this work.

We remark that our analysis is purely numerical since the solutions for $I_{1}=0$ require large values of $V_{12}$ at a given set of voltages $V_{13}$ and $V_{34}$. This fact prevents us from employing a perturbative scheme in terms of the dc voltages. The charge current through the upper dot is composed of the current directly induced by the bias $V_{12}$ and the contribution due to the charge fluctuations caused by the transport in the lower dot. The latter contribution is precisely the drag effect, which is much less significant to the creation of a charge flow through the *up* dot than the effect of a voltage directly applied between the upper terminals. The need for a numerical analysis of this system is hereby justified. To find the roots of the currents for a given set of parameters, we implemented a bisection algorithm (see Appendix A).

Since there is no magnetic field present in our system, its dynamical evolution is time-reversible. Accordingly, microreversibility ensures that the RRs should be satisfied for stalling currents far from equilibrium, as discussed in Reference [15]. In this section, we analyze both the case when the charge and heat currents stall at the same time, as well as the scenario when they do not necessarily vanish simultaneously for the same voltage configuration. The Onsager matrix for our two dot system with four leads should be of dimension 8×8 with elements denoted by $L_{ij,mn}$. In the absence of a magnetic field, Onsager’s relations imply $L_{ij,mn}=L_{ji,mn}$. Furthermore, charge conservation laws imply relations such as $I_{2}=-I_{1}$ and therefore more elements of the Onsager matrix are related. At the stalling configuration, we thus check for the fulfilment of the particular relation $L_{12,11}=L_{21,11}$, with
(24)$$L_{12,11}\equiv L_{12}=\frac{\partial I_{1}}{\partial T_{1}},\;L_{21,11}\equiv L_{21}=\frac{1}{T_{1}}\frac{\partial J_{1}}{\partial V_{1}}$$

As we can see, $I_{1}=I_{charge}$ and $J_{1}=J_{heat}$ in terms of the example in Equation (Equation 22). Here, we consider as conjugate forces the *absolute* potentials and temperatures. This is justified since the thermodynamic variables of the quantum dots do not show up in the currents, and therefore differentiating them with respect to the gradients $\Omega_{i}-\Omega_{\mathrm{dot}}$ yields the same result as differentiating with respect to $\Omega_{i}$ (where Ω represents either a voltage or a temperature). A summary of our first results is presented in Figure 3. We show the coefficients for a given $V_{12}$ as a function of $V_{13}$. It is understood that the value of $V_{34}$ at each point corresponds to the one where stalling has been numerically found. We consider four cases: (i) the globally stalled configuration depicted in Figure 3a; (ii) the locally charge-stalled case shown in Figure 3b; (iii) the locally heat-stalled scenario in Figure 3c; and (iv) a configuration where none of the currents vanish, as shown in Figure 3d. Firstly, we notice that the RRs are satisfied at the configurations where the current $I_{1}$ stalls [cases shown in Figure 3a,b] with $J_{1}$ being either zero or not. On the other hand, considering the stalling points of $J_{1}$ [see Figure 3c], in general, we do not observe an equality between $L_{12}$ and $L_{21}$. Even so, there are some exceptions (not shown here) in which the RR are satisfied despite having $I_{1}\neq 0$ and $J_{1}=0$. For these cases, however, we checked that they do not follow the FDT.

We now move on to study the validity of the fluctuation–dissipation theorem. In this case, we only consider the FDT for the charge currents. Firstly, we give explicit expressions for the relations between the transport coefficients, i.e., the FDRs. They have been established for the non-equilibrium case. Here, it is instructive to first consider the FDT *near equilibrium*. We consider the following voltage expansion of the currents around the equilibrium point $V_{i}=0$:(25)$$I_{\alpha }=\sum_{\beta }G_{\alpha ,\beta }^{eq}V_{\beta }+\sum_{\beta ,\gamma }G_{\alpha ,\beta \gamma }^{eq}V_{\beta }V_{\gamma }+O\left(V^{2}\right)$$
where the *n*th order conductances $G_{\mu ,\nu_{1}\ldots \nu_{n}}^{eq}={\left(\partial^{n}I_{\mu }/\partial V_{\nu_{1}}\ldots V_{\nu_{n}}\right)}_{V_{i}=0}$ are related to *n*th order FDRs. For instance, at second-order equilibrium FDRs lead to the FDT
(26)$$S_{\alpha \beta }^{eq}=k_{B}T\left(G_{\alpha ,\beta }^{eq}+G_{\beta ,\alpha }^{eq}\right)$$

The non-equilibrium FDT is then established to have the same form replacing the equilibrium condition by the stalling condition.

For our particular device, we investigated the non-equilibrium FDT for two cases. We tested the FDT only for the upper dot charge current, i.e., $I_{1}=-I_{2}$. The results are shown in Figure 4, where we check the FDT for the drag current, i.e.,
(27)$$S_{11}=2k_{B}TG_{1,1}$$
as well as the FDT involving the cross-correlations between the drag current ($I_{1}$) and the drive current ($I_{3}$) contributions, i.e.,
(28)$$S_{13}=k_{B}T\left(G_{1,3}+G_{3,1}\right)$$
where in both cases the noise $S_{\alpha \beta }$ was computed by applying the Full Counting Statistics (FCS) formalism described in Appendix B. We observe that only the former relation for the drag current is satisfied (Figure 4a,b) since $I_{1}$ vanishes but $I_{3}$ does not. The FDT involving cross-correlations between $I_{1}$ and $I_{2}$ also holds since both of the currents stall (not included). These results are independent of whether the heat current vanishes (see Figure 4a for $J_{1}=0$, i.e., the strong coupling regime) or not (see Figure 4b with $J_{1}\neq 0$). In the two remaining cases (Figure 4c,d), the fact that the drive current $I_{3}$ does not vanish prevents the fulfilment of the FDT for the cross-correlations $S_{13}$.

To make a complete description of the transport under stalling conditions we now discuss a remarkable result involving the third cumulants of the current. In this case, we talk about fluctuation–dissipation relations instead of the FDT. As mentioned in the Introduction, the FDRs were originally formulated by adding to the transition rates and the excess of entropy production the external potential that drives the system out of equilibrium [8]. In that sense, it is possible to establish relations between the transport coefficients such as nonlinear conductances, non-equilibrium noises, and the third cumulant. In all these cases, the transport coefficients are computed at the non-equilibrium configuration. In particular, López et al. [26] found that the following FDR is satisfied under equilibrium conditions:(29)$$C_{\alpha \beta \gamma }={\left(k_{B}T\right)}^{2}\left(G_{\alpha ,\beta \gamma }+G_{\beta ,\gamma \alpha }+G_{\gamma ,\alpha \beta }\right)$$
where $C_{\alpha \beta \gamma }=\langle I_{\alpha }I_{\beta }I_{\gamma }\rangle$ are the third-order cumulants. We ignored the indices referring to the spin degree of freedom appearing in the original paper as in our system we have spin degeneracy due to the absence of a magnetic field. Here, we checked for the fulfilment of the previous relation at stalling conditions far from equilibrium, where all the nonlinear transport coefficients $G_{\alpha ,\beta \gamma }$ are computed under non-equilibrium conditions. Note that this can be rewritten as
(30)$$C_{\alpha \beta \gamma }=3{\left(k_{B}T\right)}^{2}G_{\left(\alpha ,\beta \gamma \right)}$$
where we understand $G_{\left(\alpha ,\beta \gamma \right)}$ as the symmetrization with respect to the three indices, $G_{\left(\alpha ,\beta \gamma \right)}$ = $\left(1/3!\right)\left(G_{\alpha ,\beta \gamma }+G_{\alpha ,\gamma \beta }+G_{\beta ,\gamma \alpha }+G_{\beta ,\alpha \gamma }+G_{\gamma ,\alpha \beta }+G_{\gamma ,\beta \alpha }\right)$.

We explored the fulfilment of Equation (Equation 29) when stalling currents are present in the system, with $C_{\alpha \beta \gamma }$ again computed with help of FCS (see Appendix B). Figure 5 represents the third cumulant fluctuation relations. The case in which $I_{1}=0$ is shown in Figure 5a when $J_{1}=0$ and in Figure 5b when $J_{1}\neq 0$. In these two scenarios, the FDRs are fulfilled. However, when the cumulant relation involves currents from both the drive (either $I_{3}$ or $I_{4}$) and the drag (either $I_{1}$ or $I_{2}$) subsystems, then the corresponding FDR is no longer satisfied. Finally, for completeness, our last result is shown in Figure 6, where the FDT and third-order cumulant relations are displayed for cases where the system is not in a stalling configuration. As can be seen, none of these relations hold, as expected.

## 4. Conclusions

We provide evidence for the validity of the fluctuation–dissipation theorem and Onsager’s reciprocal relations far from equilibrium at stalling configurations where $I_{1}=0$. Additionally, we successfully tested the fluctuation relations for the third cumulant in which all the transport coefficients are calculated at stalling but far from equilibrium. The positive results are good news, as they confirm that there are indeed some situations in which a system driven far from equilibrium enjoys near-equilibrium properties, and can therefore be analyzed by means of the well-known theoretical models of equilibrium thermodynamics.

A possible extension to this work is to investigate the behavior of stalling currents and the validity of the non-equilibrium relations with transport coefficients at stalling configurations in cases where the system exhibits purely quantum effects, such as quantum transport under the preservation of phase coherence when higher-order tunneling effects are included.

## Figures and Tables

**Figure 1 entropy-22-00008-f001:**
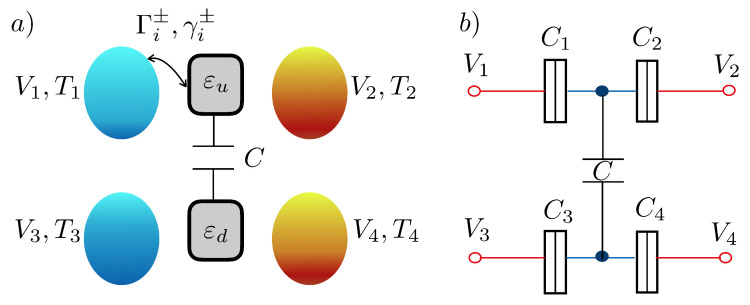
(**a**) Double quantum dot capacitively coupled to four terminals held at potentials $V_{i}$ and temperatures $T_{i}$, for i=1,2,3,4. The transition rates $\Gamma_{i}^{\pm }$ and $\gamma_{i}^{\pm }$ for each barrier are described in the main text. (**b**) Electrostatic sketch showing the capacitors and voltages involved in the description of the energy levels of the quantum dots.

**Figure 2 entropy-22-00008-f002:**
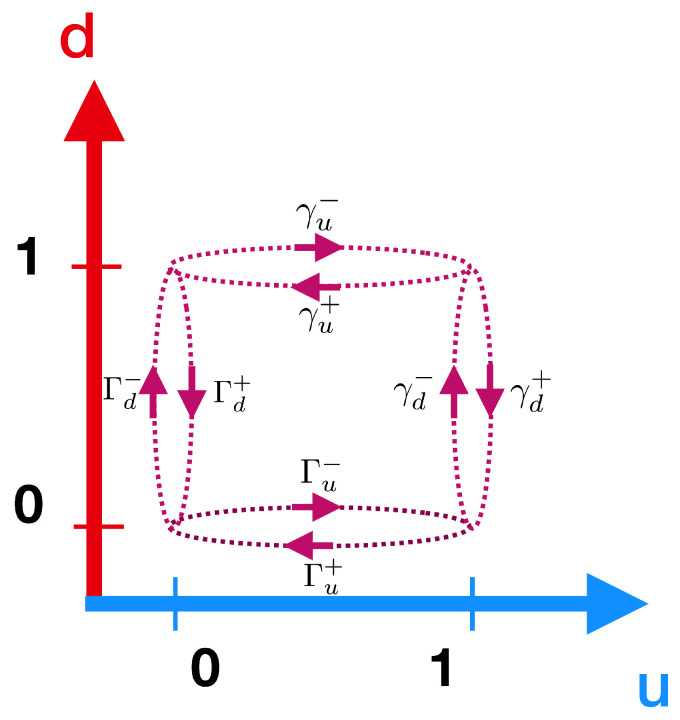
Scheme for the transition rates between the two dot states.

**Figure 3 entropy-22-00008-f003:**
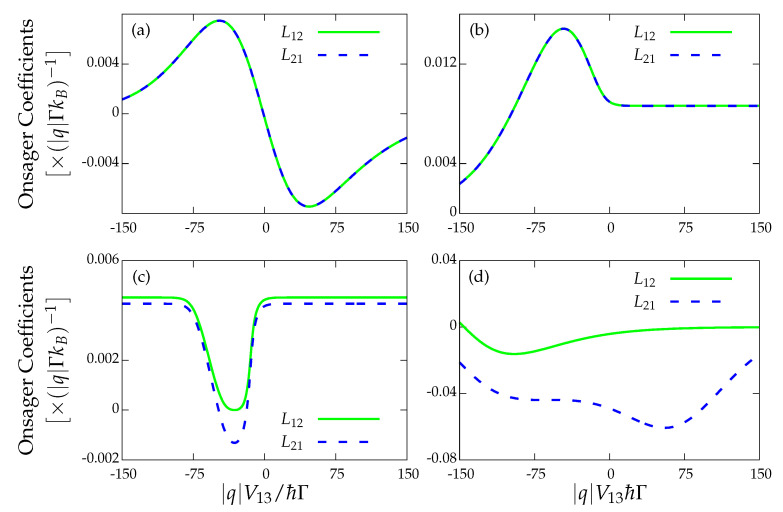
Onsager coefficients $L_{12}$ and $L_{21}$ versus the $V_{13}$ bias voltage at the indicated $V_{12}$ biases: (**a**) strong coupling configuration ($\Gamma_{2}=\gamma_{1}=0$) with $I_{1}=0$ and $J_{1}=0$; (**b**) $I_{1}=0$ and $J_{1}\neq 0$; (**c**) $I_{1}\neq 0$ and $J_{1}=0$; and (**d**) $I_{1}\neq 0$, and $J_{1}\neq 0$. $\Gamma_{i}=\gamma_{i}=\Gamma$ except for $\gamma_{1}=0.1\Gamma$, $k_{B}T=5\hbar \Gamma$, $q^{2}/C_{i}=20\hbar \Gamma$, $q^{2}/C=50\hbar \Gamma$ and $\varepsilon_{u}=\varepsilon_{d}=0$. Furthermore, we consider natural units where $\hbar =-q=k_{B}=\Gamma =1$.

**Figure 4 entropy-22-00008-f004:**
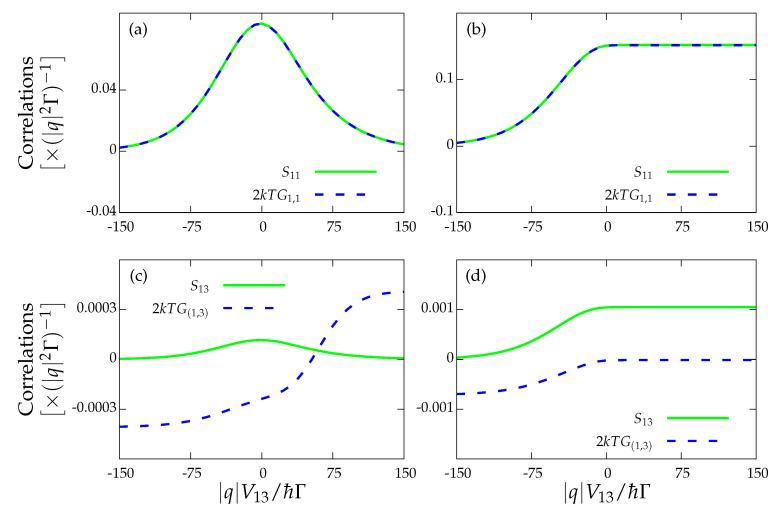
Fluctuation–dissipation theorem $S_{\alpha \beta }=2k_{B}TG_{\left(\alpha ,\beta \right)}$ for: (**a**) the strong coupling regime for the drag current, $I_{1}=0$ and $J_{1}=0$; (**b**) the locally charge-stalled configuration, $I_{1}=0$ and $J_{1}\neq 0$; and (**c**,**d**) the drag and drive currents with $I_{1}=0$ and $J_{1}=0$, and $I_{1}=0$ and $J_{1}\neq 0$, respectively. The rest of the parameters are those of Figure 3.

**Figure 5 entropy-22-00008-f005:**
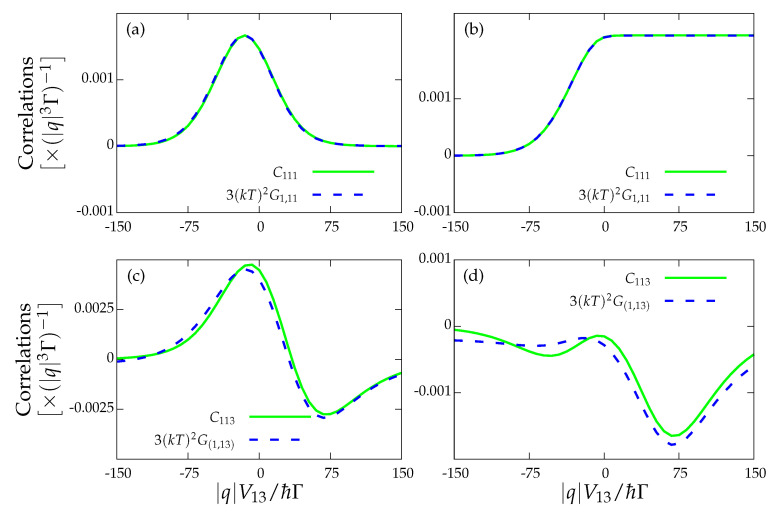
Third-order fluctuation–dissipation relations $C_{\alpha \beta \gamma }=3{\left(k_{B}T\right)}^{2}G_{\left(\alpha ,\beta \gamma \right)}$ for: (**a**) the strong coupling regime for the drag current $I_{1}=0$ and $J_{1}=0$; (**b**) the locally charge-stalled configuration $I_{1}=0$ and $J_{1}\neq 0$; and (**c**,**d**) the drag and drive currents for $I_{1}=0$ and $J_{1}=0$, and $I_{1}=0$ and $J_{1}\neq 0$, respectively. The rest of the parameters are those of Figure 3.

**Figure 6 entropy-22-00008-f006:**
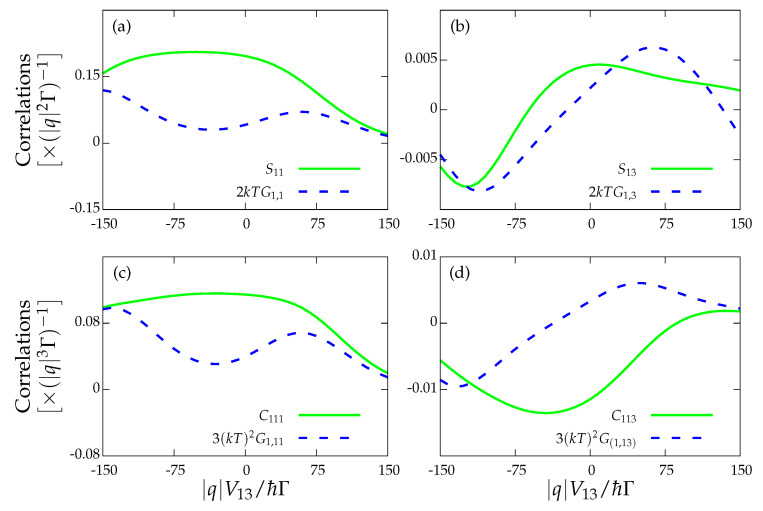
Fluctuation–dissipation theorem in a non-stalled configuration $I_{1}\neq 0$ and $J_{1}\neq 0$ for: (**a**) the drag current; and (**b**) the drag and drive currents. Fluctuation–dissipation relations in a non-stalled configuration $I_{1}\neq 0$ and $J_{1}\neq 0$ for: (**c**) the drag current; and (**d**) the drag and drive currents. The rest of the parameters are those of Figure 3.

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
