# Peer review of "Effective Equilibrium in Out-of-Equilibrium Interacting Coupled Nanoconductors"

_entropy, 2019, doi:10.3390/e22010008_

Round 1
Reviewer 1 Report
This paper studies two quantum dots coupled to each other electrostatically and they are coupled to reserviors electrostatically. Under non-equilibrium situations, some equilibrium relations of physical parameters are obeyed. The system is in the incoherent regime. the transport processes are described by the master equations. In general the paper is well written for this kind of systems. The results are useful. However, the paper can be improved in the following ways:
The term transition rates and tunneling rates are used in the modelling of systems. However, the term of electrostatically coupled are also used.Usually electrostatically coupled means there is no transition between the components and the coupling is purely via the electrostatic force. Besides, under ac conditions electrostate coupled allows current flow. the terminology used in the manuscirpt is confusing and misleading in this respect. the authors should rectify these problems with clear descriptions. It is natural to extend the study to coherent regime. Can the author provide some prediction based on physical intuition/understanding of the coherent and inchorent processes about whether similar conclusion can be obtained in the coherent regime in the conclusion? For example, it is known in resonant tunneling, the coherent picture and incoherent (used to call sequential tunneling) picture have similar mathematical expressions ( see a paper by Prof Payne of Cambridge in journal of physics). can this support that the same conclusion applies to the coherent regime. Of course the discussion is not limited to this proposed approach and can be based on the authors' own understanding.
The paper can be published after the issues are addressed.
Author Response
Dear Editor,
Below I append the answer to the Referee reports.

Reviewer 2 Report
Please find the pdf file (peer-review-5486470.v6.pdf).

Author Response

(The authors gave the same response as above.)

Reviewer 3 Report
Report on Maisel and Lopez's manuscript "Effective equilibrium in out-of-equilibrium interacting coupled nanoconductors"
This is a very interesting work, and I recommend its publication in "Entropy". However, there are a few things that are described too rapidly in the text, which makes it confusing to read (see below). A slight improvement of the manuscript would solve this without any problem.
[A] TEMPERATURES.
My critical confusion was about temperatures. Where are the temperatures are different, and where they are the same?
In Fig 1a there are four temperatures T_1, ..., T_4.
So I initially thought that the calculations were done with all leads are at different temperatures with their biases chosen ensure the condition for stalling currents. However, in section 1.1 there is only one temperature "T" in the formulas, starting from the definition of the Fermi function below Eq. (4). Then in section 1.2, there are two temperatures T and Delta T. Then in Eqs. (27-30) we are back to only one temperature T.
Upon reading the manuscript a few times, I now believe the FDT results are for uniform temperature T, while Onsager's reciprocity relations are for two temperatures T and T+Delta T,
so T_1= T+Delta T and T_2=T_3=T_4=T.
Is this correct?
If so, to avoid confusion, this should be explicitly written somewhere,
and Fig 1a should only show T and Delta T (as for Onsager's reciprocity relations), and then one can write that Delta T=0 for the part on FDTs.
[B] SIZE OF ONSAGER'S MATRIX.
The system has 4 leads with 2 currents in each lead (charge and heat current); hence the Onsager matrix in Eq (22) should be a 8-by-8 matrix. This means L_{12} and L_{21} in Eq (22) should be 4-by-4 matrices, with Onsager's reciprocity relation saying that
L_{12;ij}= L_{21;ji}
where the subscript ij indicates the ij th element of the matrix.
However the manuscript only mentions a single matrix element.
After some thought, I guessed the absence of the B-field means that
one can study Onsager's reciprocity relation for just the ij th element of the L_{12} and L_{21} matrix. This being because Onsager+time-reversal implies that
L_{12;ij}= L_{21;ij}.
Then a violation of this implies a violation of Onsager, because we believe that time-reversal is never violated at zero B-field.
Is that the correct logic?
If so, maybe the manuscript should explain this, starting from the 4-by-4 matrices for L_{12} and L_{21}, and then reduce down to the specific element of interest.
[C] MISSING DEFINITION FOR J_{charge} & J_{heat}
Eq. (22) introduces J_{charge} & J_{heat}, but does not give them in terms of the I_1 and J_1 that are used in the rest of the manuscript. I assume J_{charge}= I_1 and J_{heat}= J_1-V_1 I_1. However it would be nice to see that given explicitly to avoid confusion.
Author Response

(The authors gave the same response as above.)

Round 2
Reviewer 1 Report
the revisions are satisfactory.